# Smoothing Structured Decomposable Circuits

**Andy Shih**
University of California, Los Angeles
andyshih@cs.ucla.edu

**Guy Van den Broeck**
University of California, Los Angeles
guyvdb@cs.ucla.edu

**Paul Beame**
University of Washington
beame@cs.washington.edu

**Antoine Amarilli**
LTCI, Télécom Paris, IP Paris
antoine.amarilli@telecom-paris.fr

## Abstract

We study the task of *smoothing* a circuit, i.e., ensuring that all children of a $\oplus$-gate mention the same variables. Circuits serve as the building blocks of state-of-the-art inference algorithms on discrete probabilistic graphical models and probabilistic programs. They are also important for discrete density estimation algorithms. Many of these tasks require the input circuit to be smooth. However, smoothing has not been studied in its own right yet, and only a trivial quadratic algorithm is known. This paper studies efficient smoothing for structured decomposable circuits. We propose a near-linear time algorithm for this task and explore lower bounds for smoothing decomposable circuits, using existing results on range-sum queries. Further, for the important case of All-Marginals, we show a more efficient linear-time algorithm. We validate experimentally the performance of our methods.

## 1 Introduction

Circuits are directed acyclic graphs that are used for many logical and probabilistic inference tasks. Their structure captures the computation of reasoning algorithms. In the context of machine learning, state-of-the-art algorithms for exact and approximate inference in discrete probabilistic graphical models [Chavira and Darwiche, 2008; Kisa *et al.*, 2014; Friedman and Van den Broeck, 2018] and probabilistic programs [Fierens *et al.*, 2015; Bellodi and Riguzzi, 2013] are built on circuit compilation. In addition, learning tractable circuits is the current method of choice for discrete density estimation [Gens and Domingos, 2013; Rooshenas and Lowd, 2014; Vergari *et al.*, 2015; Liang *et al.*, 2017]. Circuits are also used to enforce logical constraints on deep neural networks [Xu *et al.*, 2018].

Most of the probabilistic inference algorithms on circuits actually require the input circuit to be *smooth* (also referred to as *complete*) [Sang *et al.*, 2005; Poon and Domingos, 2011]. The notion of smoothness was first introduced by Darwiche [2001] to ensure efficient model counting and cardinality minimization and has since been identified as essential to probabilistic inference algorithms. Yet, to the best of our knowledge, no efficient algorithm to smooth a circuit has been proposed beyond the original quadratic algorithm by Darwiche [2001].

The quadratic complexity can be a major bottleneck, since circuits in practice often have hundreds of thousands of edges when learned, and millions of edges when compiled from graphical models. As such, in the latest Dagstuhl Seminar on "Recent Trends in Knowledge Compilation", this task of smoothing a circuit was identified as a major research challenge [Darwiche *et al.*, 2017]. Therefore, a more efficient smoothing algorithm will increase the scalability of circuit-based inference algorithms.

Intuitively, smoothing a circuit amounts to filling in the missing variables under its $\oplus$-gates. In Figure 1a we see that the $\oplus$-gate does not mention the same variables on its left side and right side,

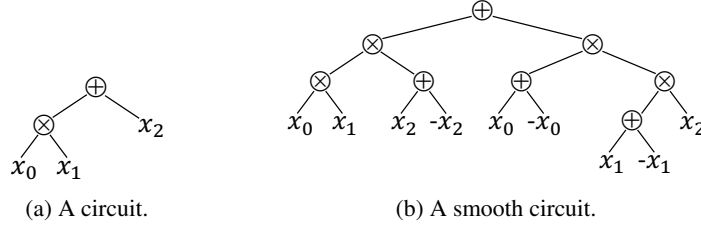

(a) A circuit.                    (b) A smooth circuit.

Figure 1: Two equivalent circuits computing $(x_0 \otimes x_1) \oplus x_2$. The left one is not smooth and the right one is smooth.

so we fill in the missing variables by adding tautological gates of the form $x_i \oplus -x_i$, resulting in the smooth circuit in Figure 1b. Filling in these missing variables is necessary for probabilistic inference tasks such as computing marginals, computing probability of evidence, sampling, and approximating Maximum A Posteriori inference [Sang *et al.*, 2005; Chavira and Darwiche, 2008; Friesen and Domingos, 2016; Friedman and Van den Broeck, 2018; Mei *et al.*, 2018]. The task of smoothing was also explored by Peharz *et al.* [2017], where they look into preserving smoothness when augmenting Sum-Product Networks for computing Most Probable Explanations.

In this paper we propose a more efficient smoothing algorithm. We focus on the commonly used class of *structured decomposable circuits*, which include structured decomposable Negation Normal Form, Sentential Decision Diagrams, and more [Pipatsrisawat and Darwiche, 2008; Darwiche, 2011]. Intuitively, structuredness requires that circuits always consider their variables in a certain way, which is formalized as a tree structure on the variables called a *vtree*.

Our first contribution (Section 4) is to show a near-linear time algorithm for smoothing such circuits, which is a clear improvement on the naive quadratic algorithm. Specifically, our algorithm runs in time proportional to the circuit size multiplied by the inverse Ackermann function $\alpha$ of the circuit size and number of variables[1] (Theorem 4.7).

Our second contribution (Section 5) is to show a lower bound of the same complexity, on smoothing decomposable circuits for the restricted class of smoothing algorithms that we call *smoothing-gate algorithms* (Theorem 5.2). Intuitively, smoothing-gate algorithms are those that retain the structure of the original circuit and can only make them smooth by adding new gates to cover the missing variables. This natural class corresponds to the example in Figure 1 and our near-linear time smoothing algorithm also falls in this class. We match its complexity and show a lower bound on the performance of *any* smoothing-gate algorithm, relying on known results in the field of range-sum queries.

Our third contribution (Section 6) is to focus on the probabilistic inference task of All-Marginals and to propose a novel linear time algorithm for this task which bypasses the need for smoothing, assuming that the weight function is always positive and supports all four elementary operations of $\oplus, \ominus, \otimes, \oslash$ (Theorem 6.1). These results are summarized in Table 1.

Our fourth contribution (Section 7) is to study how to make a circuit smooth while preserving structuredness. We show that we cannot achieve a sub-quadratic smoothing algorithm if we impose the same vtree structure on the output circuit unless the vtree has low height (Prop. 7.1).

Our final contribution (Section 8) is to experiment on smoothing and probabilistic inference tasks. We evaluate the performance of our smoothing and of our linear time All-Marginals algorithm.

The rest of the paper is structured as follows. In Section 2 we review the necessary definitions, and in Section 3 we motivate the task of smoothing in more detail. We then present each of our five contributions in order in Sections 4, 5, 6, 7 and 8. We conclude in Section 9.

## 2 Background

Let us now define the model of circuits that we study (refer again to Figure 1 for an example):

Table 1: Summary of results on structured decomposable circuits. We let $n$ be the number of variables and $m$ be the size of the circuit.

| Task | Operations | Complexity |
|------|-----------|------------|
| Smoothing | $\oplus, \otimes$ | $O(m \cdot \alpha(m, n))$ |
| Smoothing* | $\oplus, \otimes$ | $\Omega(m \cdot \alpha(m, n))^*$ |
| All-Marginal | $\oplus, \ominus, \otimes, \oslash$ | $\Theta(m)$ |

\* For *smoothing-gate algorithms* on decomposable circuits.

**Definition 2.1.** *A **logical circuit** is a rooted directed acyclic graph where leaves are literals, and internal gates perform disjunction ($\oplus$-gates) or conjunction ($\otimes$-gates). An **arithmetic circuit** is one where leaves are numeric constants or variables, and internal gates perform addition ($\oplus$-gates) or multiplication ($\otimes$-gates). The **children** of an internal gate are the gates that feed into it.*

We focus on circuits that are *decomposable* and more precisely that are *structured*. We first define decomposability:

**Definition 2.2.** *For any gate $p$, we call $vars_p$ the set of variables that appear at or below gate $p$. A circuit is **decomposable** if these sets of variables are disjoint between the two children of every $\otimes$-gate. Formally, for every $\otimes$-gate $p$ with children $c_1$ and $c_2$, we have $vars_{c_1} \cap vars_{c_2} = \emptyset$.*

We then define structuredness, by introducing the notion of a *vtree* on a set of variables:

**Definition 2.3.** *A **vtree** on a set of variables $S$ is a full binary tree whose leaves have a one-to-one correspondence with the variables in $S$. We denote the set of variables under a vtree node $p$ as $u_p$.*

**Definition 2.4.** *A circuit **respects** a vtree $V$ if each of its $\otimes$-gate has 0 or 2 inputs, and there is a mapping $\rho$ from its gates to $V$ such that:*

- *For every variable $c$, the node $\rho(c)$ is mapped to the leaf of $V$ corresponding to $c$.*

- *For every $\oplus$-gate $c$ and child $c'$ of $c$, the node $\rho(c')$ is $\rho(c)$ or a descendant of $\rho(c)$ in $V$.*

- *For every $\otimes$-gate $c$ with children $c_1, c_2$, letting $v_l$ and $v_r$ be the left and right children of $\rho(c)$, the node $\rho(c_1)$ is $v_l$ or a descendant of $v_l$ and $\rho(c_2)$ is $v_r$ or a descendant of $v_r$.*

*A circuit is **structured decomposable** if it respects some vtree $V$. The circuit is then decomposable.*

Recall that a circuit can be preprocessed in linear time to ensure that each $\otimes$-gate has 0 or 2 inputs.

Structured decomposability was introduced in the context of logical circuits, and it is also enforced in Sentential Decision Diagrams, a widely used tractable representation of Boolean functions [Darwiche, 2011]. This property allows for a polytime conjoin operation and symmetric/group queries on logical circuits [Pipatsrisawat and Darwiche, 2008; Bekker *et al.*, 2015]. For circuits that represent distributions, structured decomposability allows multiplication of these distributions [Shen *et al.*, 2016], efficient computation of the KL-divergence between two distributions [Liang and Van den Broeck, 2017], and more. Structured decomposable circuits are also used when one wants to induce distributions over arbitrary logical formulae [Kisa *et al.*, 2014] or compile a logical formula bottom-up [Oztok and Darwiche, 2015].

Next, we review another property of logical circuits that is relevant for probabilistic inference tasks [Darwiche, 2001; Choi and Darwiche, 2017].

**Definition 2.5.** *A logical circuit on variables $\mathbf{X}$ is **deterministic** if under any input $\mathbf{x}$, at most one child of each $\oplus$-gate evaluates to true.*

In the rest of this paper, we will let $n$ denote the number of variables in a circuit and let $m \geq n$ denote the size of a circuit, measured by the number of edges in the circuit.

## 3  Smoothing

We focus on the probabilistic inference tasks of weighted model counting and computing All-Marginals [Sang *et al.*, 2005; Chavira and Darwiche, 2008]. We will study weighted model counting

in the more general form of *Algebraic Model Counting* (AMC) [Kimmig *et al.*, 2016]. To describe these tasks, we define instantiations, knowledge bases and models.

**Definition 3.1.** *Given a set of variables* $\mathbf{X}$*, a full assignment of all the variables in* $\mathbf{X}$ *is called an* ***instantiation***. *A set* $f$ *of instantiations is called a* ***knowledge base***, *and each instantiation in* $f$ *is called a* ***model***.

The AMC task on a knowledge base $f$ and a weight function $w$ (a mapping from the literals to the reals) is to compute $s$ from Equation 1. The task of All-Marginals is to compute the partial derivative of $s$ with respect to the weight of each literal as in Equation 2.

$$s = \bigoplus_{\mathbf{x} \in f} \bigotimes_{x \in \mathbf{x}} w(x) \quad \text{AMC} \quad (1) \qquad\qquad \left\{ \frac{\partial s}{\partial w(x)}, \frac{\partial s}{\partial w(-x)} \,\Big|\, X \in \mathbf{X} \right\} \quad \text{All-Marginals} \quad (2)$$

On probabilistic models, $s$ is often the partition function or the probability of evidence, where the partial derivatives of these quantities correspond to all (conditional) marginals in the distribution. Computing All-Marginals efficiently significantly speeds up probabilistic inference, and is used as a subroutine in the collapsed sampling algorithm in our later experiments.

These tasks are difficult in general, unless we have a tractable representation of the knowledge base $f$. Moreover, it is important to have a smooth representation. Indeed, suppose $f$ is represented as a logical circuit that is only deterministic and decomposable but not smooth. Then, there is in general no known technique to perform the AMC and All-Marginals tasks in linear time (although there is a special case where AMC can be performed in linear time, explained below). By contrast, if $f$ is represented as a logical circuit that is deterministic, decomposable and smooth, then the AMC and All-Marginals tasks can be performed in time $O(m)$. For example, the AMC task is done by converting the deterministic, decomposable and smooth logical circuit into an arithmetic circuit, attaching the weights of the variables as numeric constants in the circuit, and then evaluating the circuit. Furthermore, when a decomposable arithmetic circuit computes a factor (a mapping from instantiations to the reals), enforcing smoothness allows it to compute factor marginals in linear time [Choi and Darwiche, 2017].

As smoothing is necessary to efficiently solve these inference tasks, we are interested in studying the complexity of smoothing a circuit. To do so, we formally define the task of smoothing.

**Definition 3.2.** *Two logical circuits on variables* $\mathbf{X}$ *are* ***equivalent*** *if they evaluate to the same output on any input* $\mathbf{x}$.

**Definition 3.3.** *A circuit is* ***smooth*** *if for every pair of children* $c_1$ *and* $c_2$ *of a* $\oplus$*-gate,* $vars_{c_1} = vars_{c_2}$.

**Definition 3.4.** *The task of* ***smoothing*** *a decomposable logical circuit is to output a smooth and decomposable logical circuit that is equivalent to the input circuit. Similarly, the task of* ***smoothing*** *a deterministic and decomposable logical circuit is to output a smooth, deterministic, and decomposable circuit that is equivalent to the input circuit.*

We only define the smoothing task over logical circuits. This is because the probabilistic inference tasks are performed by smoothing a logical circuit and then converting it into an arithmetic circuit, so it is easier for the reader to only consider smoothing on logical circuits. For the rest of the paper, we will refer to logical circuits simply as circuits. Note that we require the output smooth circuit to preserve the same properties (decomposability/determinism) as the input circuit. Indeed, there is a trivial linear time algorithm for smoothing that breaks decomposability (i.e., simply conjoin all gates with a tautological gate that mentions all variables), but then the resulting circuit may not be useful for probabilistic inference. Again, we need decomposability to compute factor marginals, and we need decomposability along with determinism to compute AMC and All-Marginals. By contrast, we do not require the output smooth circuit to be structured, because structuredness is not required to solve our tasks of AMC or All-Marginals (nevertheless, we do study structuredness in Section 7).

Sometimes, when the weight function allows division, there exists a renormalization technique that can solve AMC in linear time without smoothing the initial circuit [Kimmig *et al.*, 2016]. However, this restriction is limiting, since even if the weight function is defined over a field, division by zero may be unavoidable [Van den Broeck *et al.*, 2014]. Also, the weight function may only be defined over a semiring $(\oplus, \otimes)$ [Friesen and Domingos, 2016]. In these cases, there is no known technique to bypass smoothing. Therefore, developing an efficient smoothing algorithm is an important problem, which we address next in Sections 4 & 5.

On the other hand, one may still be interested in settings where all four elementary operations of $\oplus, \ominus, \otimes, \oslash$ on the weight function are allowed. To this end, we also propose in Section 6 a novel technique that solves the All-Marginals task in linear time when the weight function is positive, and when subtraction and division are allowed.

## 4 Smoothing Algorithm

We present our algorithm for smoothing structured decomposable circuits, based on the semigroup range-sum literature. First, we define a class of common strategies to smooth a circuit, which encompasses both the previously-known algorithm and our new algorithm.

The existing quadratic algorithm for smoothing a circuit goes to each $\oplus$-gate and inserts missing variables one by one [Darwiche, 2001]. This algorithm retains the original gates of the circuit, and adds additional gates to fill in missing variables. We will define *smoothing-gate algorithms* as the family of smoothing algorithms that retain the original gates of the circuit.

**Definition 4.1.** *Edge contraction is the process of removing each $\oplus$-gate or $\otimes$-gate with a single child, and feeding the child as input to each parent of the removed gate.*

**Definition 4.2.** *A subcircuit of a circuit is another circuit formed by taking a subset of the gates and edges of the circuit, and picking a new root. The gate subset must include the new root and all endpoints of the edge subset.*

**Definition 4.3.** *Two circuits $g$ and $h$ with gate sets $G$ and $H$ are isomorphic if there exists a bijection $B : G \to H$ between their gates such that the following conditions hold.*

1. *For any gate $p \in G$, $B(p)$ is the same type of gate as $p$.*

2. *For any gate $p_1 \in G$ and child $p_2 \in G$ of $p_1$, the gate $B(p_2)$ is a child of $B(p_1)$ in $h$.*

3. *For any gate $p_1' \in H$ and child $p_2' \in H$ of $p_1'$, the gate $B^{-1}(p_2')$ is a child of $B^{-1}(p_1')$ in $g$.*

4. *The root of $g$ maps to the root of $h$.*

*An algorithm is a **smoothing-gate algorithm** if for any edge-contracted (deterministic and) decomposable input circuit $g$, the output circuit is smooth and (deterministic and) decomposable, is equivalent to $g$, and has a subcircuit that is isomorphic to $g$ after edge contraction.*

*Smoothing-gate algorithms are intuitive, since the structure of the original circuit is preserved. This includes the quadratic algorithm, as well as algorithms which identify missing variables under each gate and attach tautological gates to fill in those missing variables, as was done in Figure 1. Formally:*

**Definition 4.4.** *A gate $g$ is called a **smoothing gate** for a set of variables $\mathbf{X}$ if $vars_g = \mathbf{X}$ and the circuit rooted at $g$ is tautological and decomposable. We denote such a gate by $SG(\mathbf{X})$.*

The structure of a smoothing gate $SG(\mathbf{X})$ is not specified. The only requirement is that it mentions all variables in $\mathbf{X}$ and is tautological and decomposable. For example, the quadratic algorithm constructs each $SG(\mathbf{X})$ by naively conjoining $x \oplus -x$ for each variable in $\mathbf{X}$ one at a time, leading to a linear amount of work per gate. In the case of structured decomposable circuits, we can do much better.

**Lemma 4.5.** *Consider a structured decomposable circuit, and let $\pi$ be the sequence of its variables written following the in-order traversal of its vtree. For any two vtree nodes $(\rho(p), \rho(c))$, we have that $u_{\rho(p)} \backslash u_{\rho(c)}$ can be written as the union of at most two intervals in $\pi$.*

*Proof.* Since $v$ is a binary tree, the in-order traversal of $v$ visits the variables of $u_{\rho(p)}$ consecutively, and the variables of $u_{\rho(c)}$ consecutively. Hence, $u_{\rho(p)}$ and $u_{\rho(c)}$ can each be written as one interval, and $u_{\rho(p)} \backslash u_{\rho(c)}$ can be written as the union of at most two intervals. $\square$

We then smooth a circuit in one bottom-up pass. If $p$ is a leaf $\otimes$-gate, replace it with $SG(u_{\rho(p)})$. If $p$ is an internal $\otimes$-gate, letting $v_l, v_r$ and $c_1, c_2$ be the children of $\rho(p)$ and $p$ respectively, replace $c_1$ with $c_1 \otimes SG(u_{v_l} \backslash u_{\rho(c_1)})$ and $c_2$ with $c_2 \otimes SG(u_{v_r} \backslash u_{\rho(c_2)})$. If $p$ is a $\oplus$-gate, replace each child $c$ with $c \otimes SG(u_{\rho(p)} \backslash u_{\rho(c)})$. By Lemma 4.5, each smoothing gate can be built by multiplying together two gates of the form $\bigotimes_{\mathbf{X}} (x \oplus -x)$, where $\mathbf{X}$ forms an interval in $\pi$. Thus, we can appeal to results from semigroup range-sums, by treating each $x \oplus -x$ as an element in a semigroup, and treating the computation of $\bigotimes_{\mathbf{X}} (x \oplus -x)$ as a "summation" in the semigroup over an interval (range).

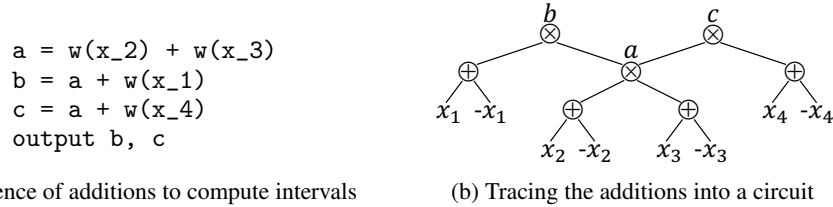

```
a = w(x_2) + w(x_3)
b = a + w(x_1)
c = a + w(x_4)
output b, c
```

(a) Sequence of additions to compute intervals

(b) Tracing the additions into a circuit

Figure 2: We construct smoothing gates for $\{x_1, x_2, x_3\}$ and $\{x_2, x_3, x_4\}$ by first passing the intervals $[1,3]$ and $[2,4]$ to the range-sum algorithm, and then tracing the sequence of additions. The trace is done by replacing $w(x_i)$ with $x_i \oplus -x_i$ and replacing each addition with a $\otimes$-gate.

**Semigroup Range-Sum.** The semigroup range-sum problem considers a sequence of $n$ variables $x_1, \ldots, x_n$, a sequence of $m \geq n$ intervals $[a_1, b_1], \ldots, [a_m, b_m]$ of these variables, and a weight function $w$ from the variables to a semigroup. The task is to compute the sum of weights of the variables in each interval, i.e. $s_j = \Sigma_{i \in [a_j, b_j]} w(x_i)$ for all $j \in [1, m]$ [Yao, 1982; Chazelle and Rosenberg, 1989]. Since $w$ is only defined over a semigroup, subtraction is not supported. That is, we cannot follow the efficient strategy of precomputing all $p_k = \Sigma_{i \in [1,k]} w(x_i)$ and outputting $s_j = p_{b_j} - p_{a_j - 1}$. Still, there is an efficient algorithm to compute all the required sums in time $O(m \cdot \alpha(m, n))$, where $\alpha$ is the inverse Ackermann function. We restate their result here.

**Theorem 4.6.** *Given $n$ variables defined over a semigroup and $m$ intervals, the sum of all intervals can be computed using $O(m \cdot \alpha(m, n))$ additions [Chazelle and Rosenberg, 1989].*

Our smoothing task can be reduced to the semigroup range-sum problem as follows. Smoothing a structured decomposable circuit of size $m$ reduces to constructing smoothing gates for $O(m)$ intervals. We pass these intervals as input to the range-sums algorithm, which will then generate a sequence of additions that computes the sum of each interval: each addition adds two numbers that are either individual variable weights or a sum that was previously computed.

We then trace this sequence of additions (see Figure 2). For the base case of $w(x_i)$, let $g(w(x_i))$ be the gate $x_i \oplus -x_i$. Then for each addition $s = t + u$, we construct a corresponding $\otimes$-gate $g(s) = g(t) \otimes g(u)$. In particular, when an addition in the sequence has computed the sum of an interval, then the corresponding gate is a smoothing gate for that interval. This process preserves determinism, so it converts a (deterministic and) structured decomposable circuit into a smooth and (deterministic and) decomposable circuit. The output circuit is generally no longer structured.

**Theorem 4.7.** *The task of smoothing a (deterministic and) structured decomposable circuit has time complexity $O(m \cdot \alpha(m, n))$, where $n$ is the number of variables and $m$ is the size of the circuit.*

Although Chazelle and Rosenberg [1989] do not formally assert a time complexity on determining the sequence of additions to perform, we show that there is no overhead to this step. That is, Chazelle and Rosenberg [1989] show that there exists a sequence of $O(m \cdot \alpha(m, n))$ additions, and we additionally prove that this sequence can be computed in time $O(m \cdot \alpha(m, n))$. The proof is in the Appendix.

## 5 Lower Bound

In this section we show a lower bound on the task of smoothing a decomposable circuit, for the family of smoothing-gate algorithms. First we state an existing lower bound on semigroup range-sums:

**Theorem 5.1.** *Given $n$ variables defined over a semigroup, for all algorithms there exists a set of $m = n$ intervals such that computing the sum of the weights of the variables for each interval takes $\Omega(m \cdot \alpha(m, n))$ number of additions [Chazelle and Rosenberg, 1989].*

We cannot immediately assert the same lower bound for the problem of smoothing decomposable circuits, for two reasons. First, we must reduce the computation of the $m$ interval sums to a smoothing problem, and express this reduction in a circuit taking no more than $O(m)$ space. Second, we must show that no smoothing algorithm is more efficient than smoothing-gate algorithms. We address the first issue but leave the second open, leading to the following theorem with the proof in the Appendix.

**Theorem 5.2.** *For smoothing-gate algorithms, the task of smoothing a decomposable circuit has space complexity $\Omega(m \cdot \alpha(m, n))$, where $n$ is the number of variables and $m$ is the size of the circuit.*

# 6 Computing All-Marginals

In this section, we focus on the specific task of computing All-Marginals on a knowledge base represented as a deterministic and structured decomposable circuit. Remember that the goal is to compute the partial derivative of the circuit with respect to the weight of each literal (Equation 2 in Section 3). If the input circuit is smooth, then we can solve the task in time linear in the size of the circuit. Therefore, with the techniques in Section 4, given an input deterministic and structured decomposable circuit, we can smooth it and then convert it into an arithmetic circuit to compute All-Marginals, all in time $O(m \cdot \alpha(m, n))$. In this section, we show a more efficient algorithm that bypasses smoothing altogether, when we assume that the weight function also supports division and subtraction and is always positive (so that we never divide by zero). The method that we propose takes time $O(m)$, which is optimal and saves us the effort of modifying the input circuit.

---

**Algorithm 1** `all-marginals`$(g, w)$

We compute partial derivatives of positive literals. The negative literals are handled similarly.

**input:** A deterministic and structured decomposable circuit $g$ on $n$ variables and a weight function $w$ that is always positive and supports $\oplus, \ominus, \otimes, \oslash$.

**output:** Partial derivatives $d_j$ for $1 \le j \le n$.

```
main(g, w):                                        8:    if p is a ⊕-gate with children C then
 1: s ← bottom-up(g, w)  // requires ⊕, ⊗, ⊘       9:       for each child k in C do
 2: return top-down(g, w, s)                       10:          l₁, r₁, l₂, r₂ ← getinterval(p, k)
                                                   11:          Δ_{l₁} ← Δ_{l₁} ⊕ D[p]
top-down(g, w, s):                                 12:          Δ_{r₁+1} ← Δ_{r₁+1} ⊖ D[p]
 1: D ← {root of g : s[root of g]}   // cache      13:          Δ_{l₂} ← Δ_{l₂} ⊕ D[p]
 2: for gates p in g, parents before children do   14:          Δ_{r₂+1} ← Δ_{r₂+1} ⊖ D[p]
 3:    if p is a leaf then d_p ← D[p]              15:          D[k] ← D[k] ⊕ D[p]
 4:    if p is a ⊗-gate with children C then       16: d₁ ← d₁ ⊕ Δ₁
 5:       m ← (⊗_{k∈C} s[k]) ⊗ D[p]                17: for i ← [2, n] do d_j ← d_{j-1} ⊕ Δ_j
 6:       for each child k in C do                 18: return d
 7:          D[k] ← D[k] ⊕ (m ⊘ s[k])
```

---

The algorithm is a form of backpropagation, and goes as follows (Algorithm 1). First, we compute the circuit output using a linear bottom-up pass over the circuit in the `bottom-up` subroutine, the details of which are omitted. During this process, we keep track of the contribution of each internal gate using the dictionary $s$. Next, we traverse the circuit top-down in order to compute the partial derivative of each gate. At a $\otimes$-gate or $\oplus$-gate, we propagate the partial derivative down to the children as needed. However, since the circuit is not smooth, there may be missing variables in the children of $\oplus$-gates, in which case the propagation is incomplete. The challenge is then to efficiently complete the propagation to the missing variables. We optimize this propagation step using range increments, which gives us the next theorem with a proof in the Appendix.

**Theorem 6.1.** *The All-Marginals task on a deterministic and structured decomposable circuit $g$ and a weight function $w$ that is always positive and supports $\oplus, \ominus, \otimes, \oslash$ has time complexity $\Theta(m)$, where $n$ is the number of variables and $m$ is the size of $g$.*

# 7 On Retaining Structuredness

Recall that our smoothing algorithm in Theorem 4.7 does not preserve structuredness of the input circuit, because it constructs smoothing gates in a way that is efficient but not structured. While structuredness is not required to solve problems such as AMC or all-marginals, it is still useful because it allows for a polytime conjoin operation, multiplication of distributions, and more (see Section 2). In this section, we show that we cannot match the performance of Theorem 4.7 while retaining structuredness, because *any* smoothing algorithm that maintains the same vtree structure must run in quadratic time. We leave open the question of whether there would be an efficient smoothing algorithm producing a circuit structured with a different vtree.

**Proposition 7.1.** *The task of smoothing a (deterministic and) structured decomposable circuit $g$ while enforcing the same vtree has space complexity $\Theta(hm)$, where $h$ is the height of the vtree and $m$ is the size of $g$.*

*Proof.* **Upper bound:** We construct smoothing gates following the structure of the vtree: for each vtree node $p$ with children $p_l$ and $p_r$, we build in constant time a structured smoothing gate for the variables that are descendants of $p$, using the smoothing gate for the variables that are descendants of $p_l$ and the one for the variables that are descendants of $p_r$. Now, we can use these gates to smooth the circuit: any interval of variables in the in-order traversal of the vtree can be written as $h$ intervals corresponding to vtree nodes, so smoothing $g$ has time complexity $O(hm)$. As with Theorem 4.6, the process of attaching smoothing gates preserves determinism.

**Lower bound:** Consider a right-linear vtree $v$ with height $h = n$ and variables $X_1, \ldots, X_n$, in that order. For simplicity, let $n$ be a multiple of 3, and consider the following functions for $y \in [0, 2^{n/3})$:

$$J_y = \bigotimes_{i=1}^{n/3} \beta(i, y) x_i \qquad\qquad K_y = \bigotimes_{i=2n/3+1}^{n} \beta(i, y) x_i$$

where $\beta(i, y) = 1$ if the $i$-th bit of the binary representation of $y$ is set, and $-1$ otherwise.

Next, consider $f = (\bigotimes_{i=1}^{n} -x_i) \oplus (\bigoplus_{y=1}^{2^{n/3}-1} (J_y \otimes K_y))$. An instantiation satisfies $f$ if all its literals are negative, or if the sign of its literals from $X_1, \ldots, X_{n/3}$ (in order) equals those from $X_{2n/3+1}, \ldots, X_n$, and are not all negative. The all-negative case is included so that $f$ mentions all $n$ variables, as otherwise $f$ would already be smooth. We can build a circuit $g$ with size $O(2^{n/3})$ that respects $v$ and computes $f$ using an Ordered Binary Decision Diagram representation [Bryant, 1986]. Yet, any smooth circuit $G$ that respects $v$ and computes $f$ has size $\Omega(n \cdot 2^{n/3})$, as we see next.

First, we use a standard notion on circuits which we refer to as a *certificate*, following the terminology by Bova *et al.* [2014]. A certificate is formed by keeping exactly one child of each $\oplus$-gate, and keeping all children of each $\otimes$-gate. Since $G$ is smooth and decomposable, every certificate of $G$ must have exactly $n$ literals, and corresponds to an instantiation of the $n$ variables. Let $C_y$ be a certificate of $G$ whose corresponding instantiation satisfies $J_y \otimes K_y$, and let $C_z$ be a certificate of $h$ whose corresponding instantiation satisfies $J_z \otimes K_z$, with $y \neq z$ and $y, z \in [1, 2^{n/3})$.

Next, let $p_i$ denote the parent of the vtree node corresponding to variable $X_i$. We will show that $C_y$ and $C_z$ cannot share a gate $w$ which maps to vtree node $p_k$ for $k \in [n/3 + 1, 2n/3]$. Suppose that such a gate $w$ exists. Then we can form a new certificate by swapping out the subtree of certificate $C_y$ rooted at $w$ with the subtree of certificate $C_z$ rooted at $w$. This new certificate now satisfies $J_y \otimes K_z$ and is a valid certificate of the circuit $G$, which contradicts the fact that $G$ computes $f$.

To finish, we consider $2^{n/3} - 1$ different certificates satisfying $J_y \otimes K_y$ for $y \in [1, 2^{n/3})$. None of these certificates can share any gates that map to vtree nodes $p_k$ for $k \in [n/3 + 1, 2n/3]$. It follows that the output circuit $G$ has size $\Omega(n \cdot 2^{n/3})$. Since the input circuit $g$ is deterministic (because it is an OBDD) and the output circuit $G$ need not be deterministic, the lower bound applies to both smoothing tasks (with and without determinism). $\square$

## 8 Experiments

We experiment on our smoothing algorithm in Section 4 and our All-Marginals algorithm in Section 6.[2] Experiments were run on a single Intel(R) Core(TM) i7-3770 CPU with 16GB of RAM.

**Smoothing Circuits.** We first study the smoothing task on structured decomposable circuits using our new smoothing algorithm (Section 4), which we compare to the naive quadratic smoothing algorithm. We construct hand-crafted circuits for which many smoothing gates are required, each of which covers a large interval. In particular, we pick $m$ large intervals $I_1, ..., I_m$ and for each interval we construct the structured gate $G_i = \bigotimes_{j \notin I_i} x_j$ for a balanced vtree. Then we take each $G_i$ and feed them into one top-level $\oplus$-gate. This triggers the worst-case quadratic behavior of the naive smoothing algorithm, while our new algorithm has near-linear behavior.

Table 2: Experiments on smoothing hand-crafted circuits and experiments on computing All-Marginals as part of the collapsed sampling algorithm. Sizes are reported in thousands (k).

(a) Time (in seconds) taken to smooth circuits.

| Size | Naive | Ours | Speedup $\times$ |
|---|---|---|---|
| 40k | $0.82 \pm 0.01$ | $0.04 \pm 0.01$ | $21 \pm 1$ |
| 416k | $50 \pm 0.3$ | $0.31 \pm 0.01$ | $161 \pm 6$ |
| 1,620k | $293 \pm 2$ | $0.74 \pm 0.04$ | $390 \pm 30$ |
| 8,500k | $6050 \pm 20$ | $4.13 \pm 0.09$ | $1470 \pm 40$ |

(b) Number of $\oplus, \ominus, \otimes, \oslash$ operations to compute All-Marginals when sampling the Segmentation-11 network.

| Size | Naive | Ours | Impr % |
|---|---|---|---|
| 100k | $28,494 \pm 598$ | $20,207 \pm 411$ | $29 \pm 3$ |
| 200k | $55,875 \pm 1,198$ | $36,101 \pm 1,522$ | $35 \pm 5$ |
| 400k | $86,886 \pm 6,330$ | $56,094 \pm 817$ | $35 \pm 6$ |

(c) Number of $\oplus, \ominus, \otimes, \oslash$ operations to compute All-Marginals when sampling the DBN-11 network.

| Size | Naive | Ours | Impr % |
|---|---|---|---|
| 100k | $172,610 \pm 1,821$ | $26,807 \pm 644$ | $84 \pm 1$ |
| 200k | $344,748 \pm 3,881$ | $51,864 \pm 851$ | $85 \pm 1$ |
| 400k | $626,235 \pm 9,985$ | $99,567 \pm 697$ | $84 \pm 1$ |

(d) Number of $\oplus, \ominus, \otimes, \oslash$ operations to compute All-Marginals when sampling the CSP-13 network.

| Size | Naive | Ours | Impr % |
|---|---|---|---|
| 100k | $36,531 \pm 1,484$ | $20,814 \pm 619$ | $43 \pm 4$ |
| 200k | $90,352 \pm 3,593$ | $38,670 \pm 1,438$ | $57 \pm 3$ |
| 400k | $122,208 \pm 9,971$ | $55,269 \pm 1,819$ | $54 \pm 6$ |

The speedup of our smoothing algorithm is captured in Table 2a. The **Size** column reports the size of the circuit. The **Naive** column reports the time taken by the quadratic smoothing algorithm, the **Ours** column reports the same value using our near-linear algorithm, and the **Speedup** column reports the relative decrease in time. The values are averaged over 4 runs.

**Collapsed Sampling.** We next benchmark our method for computing All-Marginals in Section 6 on the task of collapsed sampling, which is a technique for probabilistic inference on factor graphs. The collapsed sampling algorithm performs approximate inference on factor graphs by alternating between *knowledge compilation phases* and *sampling phases* [Friedman and Van den Broeck, 2018]. In the sampling phase, the algorithm computes All-Marginals as a subroutine.

We replace the original quadratic All-Marginals subroutine by our linear time algorithm (Algorithm 1). The requirements for Algorithm 1 are satisfied since the weight function $w$ is defined over the reals and is always positive in the experiments by Friedman and Van den Broeck [2018]. In Table 2b we report the results on the *Segmentation-11* network, which is a network from the 2006-2014 UAI Probabilistic Inference competitions. This particular network is a factor graph that was used to do image segmentation/classification (figure out what type of object each pixel corresponds to) [Forouzan, 2015]. Experiments were also performed on other networks from the inference competition, such as *DBN-11* and *CSP-13* (Table 2c & 2d). For all three networks we see a decrease in the number of $\oplus, \ominus, \otimes, \oslash$ operations needed for each All-Marginal computation. The **Size** column reports the size threshold during the knowledge compilation phase. The **Naive** column reports the number of $\oplus, \ominus, \otimes, \oslash$ operations using the original All-Marginals subroutine, the **Ours** column reports the same value using Algorithm 1, and the **Impr** column reports the relative decrease in operations. The values are averaged over 4 runs.

## 9 Conclusion

In this paper we considered the task of smoothing a circuit. Circuits are widely used for inference algorithms for discrete probabilistic graphical models, and for discrete density estimation. The input circuits are required to be smooth for many of these probabilistic inference tasks, such as Algebraic Model Counting and All-Marginals. We provided a near-linear time smoothing algorithm for structured decomposable circuits and proved a matching lower bound within the class of smoothing-gate algorithms for decomposable circuits. We introduced a technique to compute All-Marginals in linear time without smoothing the circuit, when the weight function supports division and subtraction and is always positive. We additionally showed that smoothing a circuit while maintaining the same vtree structure cannot be sub-quadratic, unless the vtree has low height. Finally, we empirically evaluated our algorithms and showed a speedup over both the existing smoothing algorithm and the existing All-Marginals algorithm.

**Acknowledgments** This work is partially supported by NSF grants #IIS-1657613, #IIS-1633857, #CCF-1837129, DARPA XAI grant #N66001-17-2-4032, NEC Research, and gifts from Intel and Facebook Research. We thank Louis Jachiet for the helpful discussion of Theorem 4.7.

## Footnotes

[1]The inverse Ackermann function $\alpha$ is defined in Tarjan [1972]. As the Ackermann function grows faster than any primitive recursive function, the function $\alpha$ grows slower than the inverse of any primitive recursive function, e.g., slower than any number of iterated logarithms of $n$.

[2]The code for our experiments can be found at `https://github.com/AndyShih12/SSDC`. There are some differences in our implementation, which we explain in the repository.

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
