[Supplementary Material · Smoothing_Appendix.pdf]



(a) Inner recursion

(b) Outer recursion

Figure 3: Recursive scheme for semigroup range-sum.

# A    Proof of Theorem 4.7

We analyze the semigroup range-sum scheme in Section 3 of Chazelle and Rosenberg [1989], and show that it can be implemented in $O(m \cdot \alpha(m, n))$ time.

The scheme, as shown in Algorithm 2, goes as follows. Let $R(t, k)$ denote the largest value of $n$ such that for any $m \geq n$, there exists an algorithm that solves the range-sum problem on $n$ variables and $m$ intervals, using $kn$ preprocessing additions and $t$ additions per interval. This gives a total of $kn + tm$ additions. We will show that $R(t, k)$ is an Ackermann function by showing the following: if $a = R(t, k-3)$, $b = R(t-2, a)$, and $n = ab$, then $R(t, k) \geq n$.

It is worth mentioning that which constants we use (2 and 3 in this case) do not matter. Any fixed constant will prove our claim. When we modify the algorithm later, it is enough to see that the algorithm works for *some* fixed constants.

For the base case we have $R(t, 1) \geq 2$ and $R(1, k) \geq 2$. To show the inductive step we first describe the preprocessing procedure (see `preprocess` in Algorithm 2). We split the $n$ variables into $b$ contiguous blocks, each of size $a$. For each single block (lines 4 & 5), we compute its prefix and suffix sums and store the results in $d$ (lines 6-12). Next, we perform an *inner recursion* where we preprocess each inner block of size $a$. Then, we treat the sum of each inner block as new variables, and perform an *outer recursion* where we preprocess the single outer block of size $b$. We store the preprocessed results from the recursive calls in $d$.

By the induction hypothesis, we know that during the preprocessing procedure, each inner block of size $a$ takes at most $a \cdot (k-3)$ additions, and the outer block of size $b$ takes at most $b \cdot a$ additions. Furthermore, computing the prefix/suffix sums for each inner block takes at most $2n$ additions. Thus, the total number of additions for preprocessing is at most $ba(k-3) + ba + 2n = (k-3)n + 3n = kn$.

Now we describe how to compute the sum of an interval using our preprocessed results. We start at the topmost level of recursion. If the interval completely falls within an inner block, then we do an inner recursion without performing any additions (Figure 3a). Otherwise, the interval straddles multiple inner blocks. Since we have computed the prefix/suffix sums for each inner block, we can shave off the edges of our interval using two additions. The remaining interval can be represented as a sum of inner blocks, which we compute by performing an outer recursion (Figure 3b).

In the case of an inner recursion, we require $t$ additions to fill the interval by induction. In the case of an outer recursion, we require 2 additions to shave off the edges of the interval, and the remaining sum can be computed using $t-2$ additions by induction.

We have shown that if $n \leq R(t, k-3)R(t-2, R(t, k-3))$, then we can solve the semigroup range-sum problem on $n$ variables and $m \geq n$ intervals using $kn$ preprocessing additions and $t$ additions per interval. Therefore $R(t, k)$ grows as fast as an Ackermann function, so the total number of required additions is $O(m \cdot \alpha(m, n))$.

In the rest of Algorithm 2, we spell out the algorithm in more detail. The subroutines `inverse-ack` and `ack` compute the necessary parameters, and `preprocess` performs the recursive preprocessing of the intervals. The subroutine `solve-interval` computes the sum of an interval using the recursive scheme described above and shown in Figure 3. The details of these subroutines are not provided.

It remains to show that we can implement this scheme without any extra time overhead. That is, can we *find* the sequence of additions to perform using $O(m \cdot \alpha(m, n))$ operations? The general strategy is to precompute a mapping from interval sizes to inner recursion depth, so that given an interval we can perform multiple consecutive inner recursions in one step.

---

**Algorithm 2** `semigroup-range-sum`$(I, w)$

---

**input:** A sequence of intervals $I = [a_1, b_1], \ldots, [a_m, b_m]$ on $n$ variables with weights $w = [w_0, \ldots, w_{n-1}]$ on the variables.

**output:** The sum of weights of variables for each interval.

`preprocess(`$\mathbf{W}, t, k$`):`

1: $a \leftarrow \texttt{ack}(t, k-3)$      $b \leftarrow \texttt{ack}(t-2, a)$
2: $d \leftarrow \emptyset$
3: **for** $i \leftarrow [0, b)$ **do**
4:    $s, e \leftarrow a \cdot i \,,\, a \cdot (i+1)$                                  // start, end of one inner block
5:    $A \leftarrow [w_s, ..., w_{e-1}]$                                                // inner block
6:    $p_s \,,\, q_{e-1} \leftarrow w_s \,,\, w_{e-1}$
7:    **for** $j$ from $s+1$ to $e-1$ **do**                          // prefix gates: walk forward
8:        $p_j \leftarrow p_{j-1} \oplus w_j$
9:        $d \leftarrow d \cup \{p_j\}$
10:    **for** $j$ from $e-2$ down to $s$ **do**                    // suffix gates: walk backward
11:        $q_j \leftarrow q_{j+1} \oplus w_j$
12:        $d \leftarrow d \cup \{q_j\}$
13:    $d \leftarrow d \cup \texttt{preprocess}(A, t, k-3)$
14:    $B_i \leftarrow p_{e-1}$                                                      // sum of inner block
15: $B \leftarrow [B_0, ..., B_{b-1}]$                                         // outer block
16: $d \leftarrow d \cup \texttt{preprocess}(B, t-2, a)$
17: **return** $d$                                                                    // preprocessed sums

`main(`$I, w$`):`

1: $n, m \leftarrow$ number of variables, number of intervals
2: $c \leftarrow \texttt{inverse-ack}(m, n)$
3: $d \leftarrow \texttt{preprocess}([w_0, ..., w_{n-1}], 2c, 3c)$
4: **for** $[a_i, b_i]$ in $I$ **do**
5:    $r_i \leftarrow \texttt{solve-interval}(a_i, b_i, 2c, 3c, d)$
6: **return** $[r_1, \ldots, r_m]$

---

The preprocessing step requires no additional overhead for finding the sequence of additions, as shown in Algorithm 2: determining which addition to perform next only takes a constant amount of time (assuming we optimize with tail recursion so we do not spend a non-constant amount of time unwinding the stack). Similarly, when we need to perform an outer recursion during the processing of one interval, we only require a constant amount of time to find the two additions (prefix and suffix pieces in Figure 3b) and call the recursion. The problem arises when we need to perform an inner recursion. Since an inner recursion does not actually performs additions, we are not allowed any time at all to find and perform the proper recursive call. So if we perform multiple consecutive inner recursions, we will end up doing a non-constant amount of work for a single addition.

As such, we will present a technique that performs multiple consecutive inner recursions, which we will call a *jump*, in a constant amount of time. After a single jump, we will either perform an outer recursion or hit a base case. In both cases, the scheme will immediately perform at least one addition, so we can absorb the (constant) cost of the jump into the addition.

### A.1 Jump Technique

Suppose we are at some level of outer recursions given by some value $t$. When we perform one inner recursion, we go from level $R(t, k)$ to level $R(t, k - c)$ for some constant $c$. When we do a jump, we need to go from level $R(t, k)$ to level $R(t, k - cj)$ for some $j \geq 1$. The details of the jumping technique are then as follows. During preprocessing, for each value of $t$ we record the sequence of block sizes $a_1^t = R(t, k-c) \geq a_2^t = R(t, k-2c) \geq \ldots \geq a_{\lfloor k/c \rfloor}^t = R(t, k - \lfloor k/c \rfloor c)$. Then, for each value $s \in [1, R(t, k)]$, we compute the smallest index $i$ such that the block size $a_i^t$ is $\leq s$. We denote this computed index $i$ by $p_s^t$. This step can be done in time $O(R(t, k) + k)$: by considering the values $s \in [1, R(t, k)]$ in decreasing order, the indices $p_s^t$ must be increasing in that order. So, we can compute all the indices with a two-pointer walk with cost $O(R(t, k) + k)$, which is negligible since

it is less than the original cost of precomputing the prefix and suffix sums at all the inner recursion levels for this outer recursion level (i.e., all choices of the value $k$ for this value of $t$).

Given an interval of size $s$ at outer recursion level $t$, we can immediately find the value $p_s^t$ and $p_s^t - 1$. We claim that it suffices to look at inner recursion levels $p_s^t$ and $p_s^t - 1$. Let $e_0 = R(t, k - p_s^t c)$ and $e_1 = R(t, k - (p_s^t - 1)c)$. By definition, we have that $e_0 \leq s \leq e_1$.

- In the case that the interval falls completely within one block of size $e_1$, we will perform an outer recursion over level $p_s^t$. We can visualize this scenario with Figure 3b, where $A$ corresponds to $e_0$ and $AB$ corresponds to $e_1$.

- Otherwise, the interval straddles exactly two blocks of size $e_1$ at the previous inner recursion level (no more than two since $e_1 \geq s$). We can express the interval as a summation of a *suffix sum over the first block* and a *prefix sum over the second block*.

In both of the above scenarios, we skip the work of performing many inner recursive calls, and jump directly to either an outer recursive call or to a base case. So for each addition, we only do a constant amount of work, and the time complexity of solving the range-sum problem on $n$ variables and $m$ intervals is $O(m \cdot \alpha(m, n))$.

## A.2 Padding

There remains one complication: the last block in a call to the `preprocess` function may have a different size from the rest of the blocks. For example, in Figure 3, if $n$ is not a multiple of $A$, then the last block will have size less than $A$. To fix this without complicating the preprocessing algorithm, we simply pad the last block so that it is the same size as all other blocks. We also make sure that when we do an outer recursion, we only pad the original blocks as opposed to padding the padded blocks from the previous recursion level. This detail ensures that the cost of padding does not compound over multiple outer recursions. Altogether, the padding technique at most doubles the memory cost of the entire algorithm.

## B Proof of Theorem 5.2

Take any set of $m$ intervals, with $m = n$. For simplicity we will let the $m$ intervals be on $n - 2$ variables (in the range $[2, n - 1]$ instead of $[1, n]$, by increasing $n$ by 2 and shifting all intervals one step to the right) so that the intervals do not touch the endpoints.

First we construct prefix gates $p_1 = x_1$ and $p_k = p_{k-1} \otimes x_k, \forall k > 1$ in a chain-like fashion, and suffix gates $s_n = x_n$, $s_k = s_{k+1} \otimes x_k, \forall k < n$ in a chain-like fashion. Then for each interval $[a_i, b_i]$, construct the gate $G_i = p_{a_i-1} \otimes s_{b_i+1}$. Next, let $g$ be the circuit $G_1 \oplus \ldots \oplus G_m \oplus p_n \oplus s_1$ (see Figure 4). We are attaching the $p_n$ and $s_1$ gate to ensure that $g$ mentions all $n$ variables, and that the smoothing-gate algorithm retains all prefix/suffix gates.

Since $g$ mentions all $n$ variables, each gate $G_i$ also needs to mention all $n$ variables to satisfy the smoothness property. By the construction of $G_i$, it is missing exactly the variables $\mathbf{X}_i = [X_{a_i}, \ldots, X_{b_i}]$. We will show that running a smoothing-gate algorithm on $g$ implicitly solves the semigroup range-sum problem on those intervals, by mapping the summation operation in the semigroup range-sum problem to the $\otimes$-gates in our circuits. Note that the circuit $g$ is indeed decomposable and edge-contracted.

Consider a smooth and decomposable circuit $h$ that is the output of running a smoothing-gate algorithm on $g$. By Definition 4.3, there exists a bijection $B$ from $g$ to a subcircuit of $h$ after edge-contraction. Let $S$ denote the graph of this subcircuit before edge-contraction: we call $S$ the *skeleton graph* (see Figure 5). We make the two following observations.

First, we consider the gates $B(G_i)$ for all $i$. In the skeleton graph $S$, there exists a path from $B(g)$ to $B(G_i)$, a path from $B(G_i)$ to $B(p_{a_i-1})$ and a path from $B(G_i)$ to $B(s_{b_i+1})$. We denote the set of gates on these paths (excluding the endpoints) over all $i$ as $T$. We observe that a gate in $T$ must have exactly one child in $S$, otherwise $S$ cannot be edge-contracted into a circuit that is isomorphic to $g$.

Since each $\oplus$-gate in $T$ has exactly one child that is in the skeleton graph $S$, we can modify $h$ by disconnecting all other children (which do not belong to $S$) from these $\oplus$-gates, and edge-contracting

Figure 4: A decomposable circuit $g$ constructed based on input intervals. Edges are solid if the parent is a $\otimes$-gate, and edges are dashed if the parent is a $\oplus$-gate. In this example the input intervals are $\{[3,3],[2,3],[2,2]\}$.

these $\oplus$-gates. We note that this operation preserves smoothness and decomposability of $h$, so each child of $B(g)$ still mentions all $n$ variables.

Second, we observe that the gate $B(p_k)$ for any $k$ cannot mention a variable outside of the range $[1,k]$. Otherwise, the circuit rooted at $B(p_n)$ would implicitly contain a $\otimes$-gate that multiplies that variable with itself, thus violating the decomposability property. A similar argument applies to the gates $B(s_k)$: they cannot mention a variable outside of the range $[k,n]$.

Let $G_i'$ denote the (unique) child of $B(g)$ that is an ancestor of $B(G_i)$. Recall that for any $i$, $G_i'$ has the gates $B(p_{a_i-1})$ and $B(s_{b_i+1})$ as descendants. Furthermore, $G_i'$ does not have any other gate in $\{B(p_j) : \forall j\} \cup \{B(s_j) : \forall j\}$ as a descendant, otherwise it would either multiply two copies of variable 1 or multiply two copies of variable $n$, and violate the decomposability property. We now remove the set of edges in $h$ that goes from some gate in $T \cup \{B(G_j) : \forall j\}$ to some gate in $\{B(p_j) : \forall j\} \cup \{B(s_j) : \forall j\}$. By the above observations, the gate $G_i'$ must now mention exactly the variables $[a_i, b_i]$. See the transition from Figure 5 to Figure 6 as an example.

We now show how to extract the variables in each interval $[a,b]$ using the following relabelling scheme to remove all remaining $\oplus$-gates. First we remove all edges leading into $G_1', \ldots, G_m'$. Each of these $m$ gates is still decomposable and smooth for the set of variables in its respective interval. Then for every $\oplus$-gate $p$ in the circuit, take one of its input wires and reroute a copy of it to each gate that $p$ feeds into. Each remaining $\otimes$-gate is now the product of one literal for each variable that was mentioned by its corresponding gate in the original circuit. These variables may be positive or negative literals, but we do not care about the polarity. We only need, for example, that if a $\otimes$-gate mentioned variables $X_1, X_3, X_5$, then it is now a product of a literal of $X_1$, a literal of $X_3$, and a literal of $X_5$.

After this operation, $G_i'$ is now exactly the product of all the variables in $[a_i, b_i]$. By setting the inputs to the circuits to be the value of the weights in the range-sum problem, and evaluating the circuits treating $\otimes$ as addition, the value to which each gate $G_i'$ evaluates is the requested sum for the $i$-th interval. So, the circuit describes a sequence of additions to compute the sum of each interval. We then apply Theorem 5.1, which implies that the bound of $\Omega(m \cdot \alpha(m,n))$ applies to the size of the output circuit $h$.

Figure 5: A smooth and decomposable circuit $h$ that is the output of a smoothing-gate algorithm on $g$. The skeleton graph $S$ is shown in red, and the set of gates $T$ are circled in blue. We proceed by taking each $\oplus$-gate in $T$ and removing their edges to children that are not in $S$. This removes the edge $(t_3, o_3)$. Next, we remove the set of edges that goes from some gate in $T \cup \{B(G_j) : \forall j\}$ to some gate in $\{B(p_j) : \forall j\} \cup \{B(s_j) : \forall j\}$. This removes the edges $(t_1, B(p_1)), (t_2, B(s_n)), (t_3, B(p_2)), (B(G_m), B(s_3))$. The gates $t_1$ and $t_3$ have no more children, so we prune them away. After this process, we get the circuit shown in Figure 6.

Figure 6: The output circuit $h$ implicitly contains a scheme for obtaining the sum of every input interval, thereby solving the semigroup range-sum problem using $O(|h|)$ additions.

## C  Proof of Theorem 6.1

Recall from Lemma 4.5 that the set of missing variables of each parent-child pair forms at most two intervals with respect to the in-order traversal of the vtree. The idea now is that propagating the partial derivative to each interval amounts to a *range increment*, i.e., increasing each variable in the interval by a constant. The naive algorithm takes quadratic time to do this for all intervals, but we can perform all range increments in linear time [Garg].

Consider an integer $n$, a set of $m$ intervals $[a_1, b_1], \ldots, [a_m, b_m]$ ($1 \leq a_i \leq b_i \leq n$), and $m$ numeric constants $c_1, \ldots, c_m$. For each integer $1 \leq j \leq n$, we wish to compute the sum $s_j = \bigoplus_{i:j\in[a_i,b_i]} c_i$. That is, if $j$ belongs to some interval $[a_i, b_i]$, then we increase $s_j$ by $c_i$. The trick is to keep track of delta variables $\delta_1, \ldots, \delta_n$. For each interval $[a_i, b_i]$, we increase $\delta_{a_i}$ by $c_i$ and decrease $\delta_{b_{i+1}}$ by $c_i$. Finally, we output $s_1 = \delta_1$ and $s_j = s_{j-1} \oplus \delta_j, j > 1$. This process, which corresponds to Lines 11-14 and 16-17 in the top-down subroutine of Algorithm 1, can be done in time $O(m)$.