[Reviews · NeurIPS 2019]

Reviewer 1



Post-rebuttal: The authors' response addresses my concerns. I don't have any further comments. ********* ORIGINALITY =========== The question of smoothing logical circuits (negative normal forms) in subquadratic time is regarded as an important open problem in the area of knowledge compilation, and this paper makes progress on it. The approach of representing the problem for structured decomposable circuits as a range sum computation problem is the main novelty of this paper. It is a simple connection but a powerful one, as it lets them use previous work in data structures to improve the running time and to prove a lower bound for smoothing algorithms. The improved algorithm for All-Marginals also uses this connection crucially. QUALITY ======= The paper is mathematically sound, and the proofs are straightforward to verify. The experimental section needs to be improved. The experiment in the "Smoothing Circuits" subsection doesn't seem interesting to me. It simply confirms the runtime analysis and is very much expected. For the "Collapsed Sampling" subsection, it plugs in their algorithm for All-Marginals into an existing algorithm for collapsed sampling. They run the modified code on the "Segmentation-11 network". No details about what the overall problem is or what the Segmentation-11 network looks like is provided. Is it a structured decomposable circuit? Are inputs for the collapsed sampling problem usually structured decomposable? CLARITY ======= The paper is reasonably well-written, and the paper is nicely organized in terms of sections. It would be better though if the authors spent more time on motivating structured decomposability and why it's important. SIGNIFICANCE ============ Structured decomposable circuits have been identified as a useful generalization of OBDD's. This work gives nearly optimal smoothing algorithms for them, which has been a longstanding open question for general logical circuits. It also shows that these circuits enjoy a fast algorithm for the all-marginals problem (provided the assumption on the weight function holds true). As an outsider, I don't know whether structured decomposable circuits are widely used in the community and so, how impactful the new smoothing algorithm will be. However, I like the fact that advances in data structures (though 30 years old!) are being brought to bear on knowledge compilation.

Reviewer 2



CLARITY The text is nicely written, understandable, and up to a number of minor issues (see details) clear. QUALITY I didn't find major flaws in the claims. Some details are listed below. Experiments are performed on hand-crafted circuits. These may not be representative for practical circuits (see also comment on significance). It may be better to report "speedup factor" rather than "Improve %" (which is near 99% always here). ORIGINALITY To the best of my non-expert knowledge the contribution seems to be original. SIGNIFICANCE The paper motivates the problem by arguing that the algorithm of Darwich (2001) is too expensive. Still, Darwich (2001), which focuses on d-DNNFs rather than general circuits) states that making a circuit smooth is reasonable for circuits occuring in practice, in particular it says "These operations preserve both the decomposability and determinism of a DNNF. They may increase the size of given DNNF but only by a factor of O(n), where n is the number of atoms in the DNNF. This increase is quite minimal in practice though.". It may be useful to clarify this difference in opinion on whether this smoothing is problematics. Another issue is that the proposed algorithm requires a structured circuit as input but doesn't preserve structuredness. Line 266 lists some tasks which don't need structuredness, but doesn't discuss how much other common tasks depend on structuredness. If only tasks not needing structuredness should be performed, then getting the circuit into a structured form (to be able to apply the proposed algorithmà may come with an additional cost. REPRODUCIBILITY The checklist asserts all information on the used data is provided, but the paper doesn't give details. The supplementary material is a zip file, but most directories seem to contain code. L 120 : please specify whether the m in O(m) gives the number of variables or the number of literal or ... If you intend to use m as a global variable throughout the paper, then please make this very clear at the beginning of the section. My guess is here that m is the number of literals. (e.g., hiding the specification of m and n in the caption of Table 1 is not sufficient) * L 159 : the definition suggests that a smoothing gate algorithm could output a circuit which is not logically equivalent to the input circuit. Is that the intention? * L 177 : it seems that for the internal x-gate, c1 and c2 could encode non-trivial circuits, and _replacing_ them with SG(u_{vl}\setminus u_{\rho(c_1)}) and SG(u_{vr}\setminus u_{\rho(c_2)}) gives a non-equivalent circuit. Maybe you want to add children to the gate? * L 181 : in contrast to L 120, it seems here m is the number of intervals (hence L 120 needed a specification of the meaning of m). * L 186 : what is the meaning of n here? * L 204 : thm 4 is ill-formulated. It says "there exists intervals such that .... takes ... time". However, once we know the intervals we can pre-compute the sums. I guess you need to say "for all algorithms, there exist m intervals such that ... takes ... time." * L 211: please define accurately "the class of smoothing-gate algorithms," * L 253 : here, m and n are nicely defined. ------------------- I appreciate the informative author feedback.

Reviewer 3



# Summary The paper proposes an efficient method, near-linear time algorithm, for smoothing a circuit, focusing on structured decomposable circuits, thus improving the naive quadratic algorithm. # Quality The authors analytically prove the complexity of the proposed algorithm in Theorem 3. The the prove the lower bound of the same complexity. Due to the result reported in Lemma 1 the authors show how to apply the semigroup range-sum leading to a method for smoothing the AC with a low time complexity (see Theorem 3). Another linear time algorithm is proposed for the All-Marginals inference task avoiding the need of smoothing. A final experimental evaluation prove the validity of the proposed methods. # Clarity The paper is well written, the concepts are well defined and the proofs correctly prove the reported statements. A suggestion could be to better explain the backgrouds on AC to make the paper more self-contained. Definition 8 could be splitted into two definitions, one for smoothness and the other for the task of smoothing. # Originality The idea is quite very simple, however the reported results in terms of theorems and experiments prove the validity of the proposed approach. # Significance The reported approach presented in the paper make able to solve inference tasks in linear time on non smoothed AC. This is a very important result.

[Author Response · NeurIPS 2019]

Thank you all for the helpful reviews. We first address concerns shared among reviewers:

**Experiments (real world networks)**: The Segmentation_11 network is a real-world network taken from the UAI Prob-
abilistic Inference competition (2006 to 2014). It is a factor graph that was used to do image segmentation/classification,
"and the goal is to figure out what type of object each pixel corresponds to" [Forouzan, 2015].

As suggested, we will run and report experiments on more networks, for a more comprehensive picture of our algorithm.

**Structured decomposability (significance)**: As suggested, we will work on motivating this. Structured decompos-
ability makes the following tasks tractable: multiplying distributions, computing KL-divergence of distributions,
equivalence checking on logical circuits, conjoining/disjoining logical circuits.

Therefore, structured decomposable ACs are used when one wants to compose a tractable representation of a distri-
bution or a logical formula using multiply/conjoin/disjoin. This includes inducing distributions over arbitrary logical
formulae [Kisa *et al.*, 2014] or compiling a logical formula bottom-up [Oztok and Darwiche, 2015].

**Reviewer 1**: *Smoothing circuits subsection...* We will focus more on the real-world networks.

*Segmentation-11 details...* See above section on **experiments**. The collapsed sampling algorithm does inference on a
factor graph by first compiling it into an SDD (a subset of structured decomposable circuit) and then smoothing the
SDD. So, the AC given as input to the smoothing task is always structured decomposable.

*Motivating structured decomposability...* See above section on **structured decomposability**.

**Reviewer 2**: *Hand-crafted not representative...* See above section on **experiments**.

*Report speedup factor...* OK, we will change this.

*Quadratic is too expensive...* In recent years the size of AC's have grown to 100k/1m, and can have hundreds/thousands
of variables [Friedman and Van den Broeck, 2018; Rooshenas and Lowd, 2014] (Seg_11_processed has $> 3k$ variables).

*Proposed algorithm requires a structured circuit...* See above section on **structured decomposability**. ACs may be
constructed through a series of multiply/conjoins/disjoins, so structuredness would already be there.

*The checklist asserts...* Sorry this wasn't clear. The Seg_11_processed network is in the folder Collapsed-
Compilation/Segmentation_11_processed/. We run inference on the given network, so there is no data to report
(which is why we checked the box originally).

*L 120* Variable $m$ is the size (number of edges) of the circuit. L98-99 was meant to specify this.

*L 159* You're right, it is better to define it to solve the smoothing task. We will update this.

*L 177* Yes, this was a typo. A fix was included in the supplemental zip file (since they already locked changes to pdf)

*L 181* Yes, here $m$ is the number of intervals. It corresponds (with a constant factor) to the size of the circuit.

*L 186* Here $n$ is the number of variables. The inv-ack takes both number of variables and number of intervals as input.

*L 204* Good point, we will re-formulate it as you suggested. Thanks.

*L 211* Ok. It is meant to refer to algorithms satisfying Def 10. We will make this more precise.

*L 253* Thank you.

*discuss more clearly the significance... improve experiments...* OK, we will work on these (see above sections).

**Reviewer 3:** *Better explain the backgrounds on AC* Ok.

*Definition 8 could be splitted...* Ok, we will update this.

*Results on real world...* See section on **experiments**.

# References

Sholeh Forouzan. Approximate inference in graphical models. *UC Irvine*, 2015.

Tal Friedman and Guy Van den Broeck. Approximate knowledge compilation by online collapsed importance sampling. In *NeurIPS*,
pages 8024–8034, 2018.

Doga Kisa, Guy Van den Broeck, Arthur Choi, and Adnan Darwiche. Probabilistic sentential decision diagrams. In *KR*, 2014.

Umut Oztok and Adnan Darwiche. A top-down compiler for sentential decision diagrams. In *IJCAI*, 2015.

Amirmohammad Rooshenas and Daniel Lowd. Learning sum-product networks with direct and indirect variable interactions. In
*ICML*, pages 710–718, 2014.


[Meta-Review · NeurIPS 2019]

The paper presents the smoothing-gate algorithm, a a near-linear time algorithm for smoothing structured decomposable logical circuits. The only downside is that the experiments are done on hand-crafted circuits only. However, the reviewers fully agree that this is an interesting and useful research direction. I fully agree. A very nice direction. Well developed and presented.